# Synthesis, Structural and Magnetic Properties of Cobalt-Doped GaN Nanowires on Si by Atmospheric Pressure Chemical Vapor Deposition

**DOI:** 10.3390/ma16010097

**Published:** 2022-12-22

**Authors:** Zhe Chuan Feng, Yu-Lun Liu, Jeffrey Yiin, Li-Chyong Chen, Kuei-Hsien Chen, Benjamin Klein, Ian T. Ferguson

**Affiliations:** 1Southern Polytechnic College of Engineering and Engineering Technology, Kennesaw State University, Marietta, GA 30060, USA; 2Department of Electrical Engineering, Graduate Institute of Photonics and Optoelectronics, National Taiwan University, Taipei 10617, Taiwan; 3National Taiwan University Center for Condensed Matter Sciences, Taipei 10617, Taiwan; 4Institute of Atomic and Molecular Sciences, Academia Sinica, Taipei 10617, Taiwan

**Keywords:** cobalt-doped GaN nanowires, atmospheric pressure chemical vapor deposition, scanning and transmission electron microscopy, X-ray diffraction, energy dispersive X-ray spectroscopy, superconducting quantum interference device

## Abstract

GaN nanowires (NWs) grown on silicon via atmospheric pressure chemical vapor deposition were doped with Cobalt (Co) by ion implantation, with a high dose concentration of 4 × 10^16^ cm^−2^, corresponding to an average atomic percentage of ~3.85%, and annealed after the implantation. Co-doped GaN showed optimum structural properties when annealed at 700 °C for 6 min in NH_3_ ambience. From scanning electron microscopy, X-ray diffraction, high resolution transmission electron microscope, and energy dispersive X-ray spectroscopy measurements and analyses, the single crystalline nature of Co-GaN NWs was identified. Slight expansion in the lattice constant of Co-GaN NWs due to the implantation-induced stress effect was observed, which was recovered by thermal annealing. Co-GaN NWs exhibited ferromagnetism as per the superconducting quantum interference device (SQUID) measurement. Hysteretic curves with Hc (coercivity) of 502.5 Oe at 5 K and 201.3 Oe at 300 K were obtained. Applied with a magnetic field of 100 Oe, the transition point between paramagnetic property and ferromagnetic property was determined at 332 K. Interesting structural and conducive magnetic properties show the potential of Co-doped GaN nanowires for the next optoelectronic, electronic, spintronic, sensing, optical, and related applications.

## 1. Introduction

Great breakthroughs on gallium nitride (GaN)-based material and devices as well as industrial applications have been of recent interest [1,2]. GaN-based one dimensional (1D) nanostructure, such as GaN nanowire, etc., has attracted much research attention [3,4,5,6,7,8,9,10,11,12,13,14,15,16,17,18,19,20,21,22]. These types of GaN-based nanowires (NWs) possess unique electronic, optical, magnetic, and catalytic features, and exhibit numerous novel and interesting properties [3,4,5,6,7,8,9,10,11,12,13,14,15,16,17,18,19,20,21,22,23,24,25,26,27,28,29]. Compared to conventional planar films, NWs, with characteristics of a high surface-to-volume ratio and uniaxial charge transport path, provide a better choice for multiple nanoscale applications. GaN NWs possess a semi-discrete density of states with a continuous transport path for carriers, due to its 1D configuration [3].

GaN NWs are good candidates for photocatalytic applications. Their energy positions and band edges are aligned with redox levels in electrolytes, leading to an improved and controllable photocatalytic activity [7]. The large surface-to-volume ratio of GaN NWs allows a high integration density on devices and systems. An enhanced light extraction efficiency and light out-coupling could be achieved in GaN NWs in comparison with the planar counterparts [6].

GaN NWs can be prepared on low-cost substrates, because nanowires possess properties less prone to dislocations and strain. NWs with small strains could have a smaller polarization field leading to a large electron-hole wave function overlap [9]. GaN NWs on Si (111) showed a better electron spin relaxation as compared to bulk GaN, because electrons were limited in quasi one-dimensional semiconductor channels [8].

III-nitride NWs with features of high radiative recombination rates, small Auger effects, high electron mobility, and tunable energy band gaps from near infrared (InN, 0.64 eV) to ultraviolet (AlN, 6.2 eV) could lead to future applications for light-emitting diodes (LEDs), photodetectors, and lasers [6,19]. NWs’ 1D nature is beneficial for epitaxial growth on dissimilar substrates (such as glass), because of the reduced strain from lattice mismatches and different thermal expansion coefficients [14]. 

GaN NWs are also beneficial for new optoelectronic and photonic devices, such as flexible LEDs, single photon sources or even photons, and plasmon and polariton nano-lasers. One-dimensional NW-sensors have been employed to detect low concentration gas species [15]. GaN NWs can be applied to improve the performance of field emission scanning probe lithography (FE-SPL) in the AFM-in-SEM probe lithography instrument [10].

GaN NWs are favorable for p-doping and achieving dislocation-free growth on a variety of lattice-mismatched substrates like Si [24,26,27] or other nonconventional substrates for GaN [16]. AlGaN NWs on Si with AlN buffer provide a good internal quantum efficiency (IQE) for deep ultraviolet (DUV) LEDs [17].

GaN NWs are suitable for gas-sensing applications, to sensitively detect noxious gases such as NO_2_, SO_2_, and so on [18]. GaN NWs on Si have been used for the detection of SO_2_ gas [20]. GaN NWs with good PL response in acidic and related solutions could be applied to measure pH and bias response with potential applications in harsh chemical environments [22].

In the field of electronic device application, the emission properties of GaN nanowires (NWs) can be improved with a larger current density and a lower applied voltage. This is attractive for vertical GaN NW vacuum field emission diodes (VFEDs) and GaN NW vacuum field emission transistors (VFETs). Devices possessing enhanced FE characteristics with low turn-on voltage, high doping concentration, small diameter, and increased height could be achieved [19].

GaN NWs have promising structures for applications of light-emitting diodes (LEDs), laser diodes (LDs), solar cells, and photocatalysts [6,13,20,25,26,27]. One-dimensional (1D) semiconductor nanowire structure is recognized as a core technology that can overcome the limitations of existing technologies in optoelectronics, and the display, bio, and environmental fields [21]. In comparison with bulk material, one-dimensional nanowires exhibit good lattice relaxation and can be grown with a low level of defects. Long-length nanowires provide larger surfaces, which could be used for sensor and chemistry applications [22].

Intense research on GaN NWs is currently being enhanced [23,24,25,26,27,28,29], with good research accomplishments such as photoelectrochemical water splitting using GaN NWs [23], selective area epitaxy of GaN NWs on Si for microsphere lithography [24], N-polar InGaN/GaN NWs for red-emitting micro-LEDs [25], InGaN/GaN NW LEDs on microsphere-lithography-patterned Si [26], GaN NWs/Si photocathodes for CO_2_ reduction [27], bending strain effects of GaN NWs [28], GaN NWs with excellent UV luminescence [29], and so on.

GaN NWs doped with 3*d*-transition metal ions (Mn, Fe, Cr, Co, and Ni) are attractive dilute magnetic semiconductors (DMSs). Partially filled *d*-states have unpaired electrons that can introduce spin properties in GaN NWs. This spin due to the hybridization of the magnetic impurity in *s*- and *p*-states of the semiconductor, is beneficial for spintronic applications. In comparison with III-V semiconductors of GaAs, InAs, and InP, the transition metal (TM)-doped GaN possesses a Curie temperature (*Tc*) beyond RT. However, it is still controversial and the origin of ferromagnetism in these semiconductors is not well understood [30].

In Co-doped GaN, it has been indicted that Co atoms are located at Ga sites of the GaN lattice and cause part of the ferromagnetism at RT [31]. Co does not show site preference in GaN crystal, while in GaN NW, Co substitutes preferably surface sites [32]. Gd-doped GaN (Gd:GaN) and Cu–Gd co-doped GaN NWs are also promising for RT ferromagnetism and spintronic applications [33]. Co-GaN NWs with a diameter of 60–200 nm and length in microns did not change the GaN wurtzite structure and did not add any secondary phases (Ga_2_O_3_ or Co_2_O_3_); it showed magnetism that is directly dependent on Co content [34]. Co-doped GaN NWs can be used for optoelectronics and spintronic devices. 

Co-doped GaN NWs have shown attractive structural, magnetic, and optical properties recently. Based on Raman analysis, Co dopant in GaN NW causes a strong electron-phonon coupling with the (LO) mode, leading to the enhancement and up-shift of LO-phonon [35]. GaN NWs exhibit the ferromagnetic (FM) coupling between the neighboring TM ions, which could be responsible for the ferromagnetism. First principal calculations show that TM dopants are able to incorporate on the outermost surface of GaN NWs [36]. The polarization of 3d electrons of TM atoms and 2p electrons of N atoms could induce the magnetic moments. Mn-doped GaN NWs with 100% spin polarization characteristics could be good candidates in spin-polarized electron photocathode applications.

Co-doped ZnO NWs, like Co-doped GaN NWs, have also attracted much research attention for spintronics and other applications [37,38,39]. TM dopants may extend the optical absorption edge and move the semiconductor density of states (DOS) to near its Fermi level of TMs. Additionally, d and f electrons in TMs cause a strong spin−orbit interaction, producing spin-polarized semiconductor bands [37]. The radius of the Co^++^ ion (0.72 Å) is very close to that of the Zn^++^ ion (0.74 Å), which makes them a unique material combination. In ZnO, Co is predicted to form a single-phase Co^++^ form and substitute the Zn^++^ position [38]. Ga and Zn are close within the periodic table. It is reasonable to expect similar behavior for Co^++^ and Ga^+++^.

Briefly, the literature up-to-now has shown that transition metal Co- and TM-doped GaN NWs can exhibit dilute magnetism with potential applications in spintronics and other fields [30,31,32,33,34,35,36]. However, a systematic investigation on the magnetic behavior of Co-GaN NWs is essential to realize these applications. 

This work is focused on a systematic investigation of the structural and magnetic properties of Co-GaN NWs. The carriers spin and conductive properties in the Co-GaN NWs are discussed. Experimental samples of Co-doped GaN NWs were grown on Si by atmospheric pressure chemical vapor deposition (APCVD) using a resistive heated furnace system. Co-doping was achieved using ion implantation. Samples were annealed in a thermal furnace to reduce possible defects. A variety of characterization techniques, including scanning electron microscopy (SEM), X-ray diffraction (XRD), photoluminescence (PL), high resolution transmission electron microscope (HR-TEM), and energy dispersive X-ray spectroscopy (EDX) were used for measurements and analyses, from which the single crystalline nature of cobalt-doped gallium nitride nanowires was identified.

Surface morphology from SEM of Co-doped GaN nanowires shows that the wires bent and the surface was damaged if the samples were not properly thermal-annealed. XRD scans show no secondary phase formation after ion implantation. A slight expansion in the lattice along the a-axis is observed. This phenomenon is ascribed to an implantation-induced stress within the nanowires. After thermal annealing, the structure resumes the original crystallinity as revealed from XRD data. HRTEM measurements and EDX atomic analysis on the surface show no clusters. A SQUID system was used for magnetic measurement, and a hysteretic curve was obtained at 300 K. Our results indicate that the cobalt-doped gallium nitride nanowires possess interesting RT ferromagnetic properties.

## 2. Experiment

### 2.1. Atmospheric Pressure Chemical Vapor Deposition (APCVD) of GaN NWs

In this work, GaN NWs were prepared using atmospheric pressure chemical vapor deposition (APCVD) based on the catalytic enhanced vapor–liquid–solid (VLS) growth mechanism. Ga and Ga_2_O_3_ powders were mixed in a vessel and served as the source materials. After placing the gold-patterned silicon substrate into the quartz tube of the APCVD system, a base pressure of 100 mTorr and an Ar gas flow rate at 80 sccm were set. Temperature in the growth chamber was gradually raised to 920 °C at 30 min before initiating the VLS process of GaN NW growth. We then switched on the NH_3_ precursor with a flow rate of 10 sccm at 920 °C for 1 h with N_2_ for carrier gas.

The VLS of GaN NWs could be elaborated. Ga metal and Ga_2_O_3_ were mixed in the vessel. Ga atoms reacted with Ga_2_O_3_ to produce Ga_2_O, which, via carrier gas, was transported to the deposition zone. Ga_2_O reacted with N at sites of gold-patterned Si substrates in the deposition zone via the reaction: 3Ga_2_O(g) + 4N → 4GaN(s) + Ga_2_O_3_(g). Several parameters such as gas flow rates, temperature, pressures, and group V to III ratio can be used to handle the growth of NWs. The effects of these parameters have been investigated and optimized for the growth of various nanostructures with a suitable growth mechanism. This is demonstrated in Section 3.1 in detail.

After the growth, NH_3_ precursor and N_2_ carrier gas flows are stopped, and the temperature is gradually decreased to RT. Using SEM examination, formation of GaN nanowires is observed, as shown in Figure 1 of Section 3.1, where the surface morphology of as-grown GaN NWs is displayed with two magnification factors of ×10 K and ×100 K.

### 2.2. Ion Implantation

A 9SDH-II tandem accelerator, with a high terminal voltage of 3 MV, at National Tsing Hua University, was used for the Co implantation. In this process, the Co atoms were placed in the chamber, and an acceleration of 72 keV was achieved using appropriate negative electrodes. Theoretical simulation by a program of TRIM (Transport of Ions in Matter) was performed, which is described in Section 3.2. Through ion implantation, selective-area p-type doping can be realized [40]. Selective area doping in GaN enables planar process technology, and to avoid the complications from the etch/regrowth process, with the subsequent annealing to activate dopant species and repair the damage to a crystal [41]. In rare earth (RE) ion-implanted GaN nanowires, the NW core is of high crystalline quality with the extended defect concentration lower than that of ion-implanted thin films. Additionally, the implantation-caused strain in NWs is efficiently relaxed and the deformation is also below that of thin films implanted under the same conditions [42]. The cobalt ion implantation at a high-fluence (5 × 10^16^ cm^−2^) into n-GaN epilayer on sapphire was studied previously [43]. When the accelerating cobalt ions hit into the GaN nanowires, it would induce collision until the bombardment energy is reduced to zero. During the implantation, the dose concentration and ion current could be controlled and adjusted. This is based on the simulation of the mechanism process of the incident ion, and it helps to control the distribution of the bombardment.

In the present work, according to the simulation for 72 keV, Co ions were distributed as a Gaussian distribution. The peak was at about 51.5 nm with a covering range of about 100 nm. Average atomic percentage was about 3.85% for a dose concentration of 4 × 10^16^ cm^−2^.

### 2.3. Annealing

After ion implantation, annealing was used to recover the structure’s crystallinity. A rapid thermal annealing (RTA) furnace was used for the annealing.

Four RTA temperatures of 500 °C, 600 °C, 700 °C, and 800 °C were first tested for optimizing a suitable temperature parameter. NH_3_ ambient was used, and annealing duration was 6 min. Nitrogen vacancy is a common defect in GaN material, so NH_3_ ambience was used for preventing nitrogen vacancies. Based upon the photoluminescence (PL) and SEM measurements on four samples annealed from lowest temperature (500 °C) to highest temperature (800 °C), the best conditions for annealing temperature were obtained (see Section 3.3).

### 2.4. Characterization Techniques

A series of techniques including SEM, XRD, PL, HRTEM, EDX, and SQUID were applied for characterization in this work. XRD measurements were done using a Bruker D8 Advance XRD (Billerica, MA, USA). PL spectra were measured by using a Renishaw micro-Raman system with the 325 nm excitation from a He-Cd laser. SEM measurements were conducted using a JEOL JSM-6700F (Akishima, Japan). HR-TEM measurements were performed using a JOEL JEM-2010F under an accelerating voltage of 200 kV and with a magnification of 1.5 M. A SQUID equipment MPMS-XL7 was employed for magnetic experiments. 

## 3. Results and Discussion

### 3.1. As Grown Undoped GaN NWs and Cobalt Ion Implantation

GaN NWs were grown on gold-coated Si substrate. After the gold-coated silicon substrate was put into APCVD quartz tube, three process stages of alloying, nucleation, and axial growth were followed.

(i) Alloying process: Au clusters stayed in the solid state up to 920 °C without the Ga vapor condensation. When the Ga vapor condensation and dissolution appeared, Ga and Au formed an alloy and liquefied. Due to the dilution of Au with Ga, the alloy droplets increased, and the elemental contrast decreased as the alloy composition changed sequentially. 

(ii) Nucleation: After the composition of alloy crossed the second liquidus line, NW nucleation starts at this step, indicating that the nucleation occurred in a supersaturated alloy liquid. 

(iii) Axial growth: After the liquid droplet was supersaturated, the target material condensed and formed nucleation sites. Further condensation/dissolution in the system then increased the amount of target material precipitation from the alloy. The material is diffused to condense at the formed liquid/solid interface, because less energy is required for crystal growth in comparison with the case of creating another nucleation site. The secondary nucleation is suppressed, and no new solid/liquid interface is created. The interface is pushed forward (or backward) to form a NW.

Growth parameters varied included (1) carrier gas, (2) growth temperature, and (3) growth time. These experiments were tested by changing one parameter while keeping the other parameters constant. 

Effects of different carrier gas were investigated using SEM; GaN nanowires images with different length and structures by changing the carrier gas were obtained. GaN NWs grown using nitrogen carrier gas seemed to be shorter than when using Argon gas. GaN NWs were not formed effectively using nitrogen carrier gas. Dissociation of nitrogen suppresses the gallium source from forming gallium nitride nanowires. GaN NWs grown using argon gas showed improved structural and morphological properties compared to using nitrogen gas. Argon gas is a common noble gas and is stable to reaction at high temperatures. Argon has the same solubility in water as oxygen gas and is 2.5 times more soluble in water than in nitrogen gas. The stable Ar element is colorless, odorless, tasteless, and nontoxic in both its liquid and gaseous forms. It is inert under most conditions and does not form stable compounds at RT. Argon was chosen to be the carrier gas in the next sets of experiments.

Effects of growth temperatures were also examined. At 700 °C, the wires rolled up; the structure seemed to have worse quality. At an increased growth temperature of 950 °C, one-dimensional zigzag nanostructures were formed. This result is like reference [44]. Growth temperature was optimized at 920 °C.

Different growth times were also used and their effects on GaN NWs formation was studied. The density of nanowires increased with the growth time. If the nanowires density was too high, the bottom of the nanowires would form a nucleation layer; this factor would affect the nanowires’ morphology and crystallinity. Hence, 1 h was chosen as the growth duration for GaN NWs and the formation of a nucleation GaN layer was prevented. Appropriate APCVD growth key parameters are listed as (Table 1): 

These parameters were used to grow the GaN nanowires in Figure 1a,b. From the SEM image, the diameter of the GaN NWs is about 40~120 nm, with an average width of about 80 nm, and the length of NWs ranged from 3 to 5 μm.

Cobalt ion implantation was performed on as-grown GaN NWs. TRIM is the most comprehensive program included for compound materials and is used in this study. Considering the diameter of nanowires as 40~120 nm, 72 keV was used as the implantation energy. Figure 1c is the result of simulation over 1,000,000 (1 million) ions. According to the simulation for 72 keV, the cobalt ions distributed following a Gaussian distribution. The peak was at about 51.5 nm, and the covering range was about 100 nm. The average atomic percentage was about 3.85% for a dose concentration of 4 × 10^16^ cm^−2^.

**Figure 1 materials-16-00097-f001:**
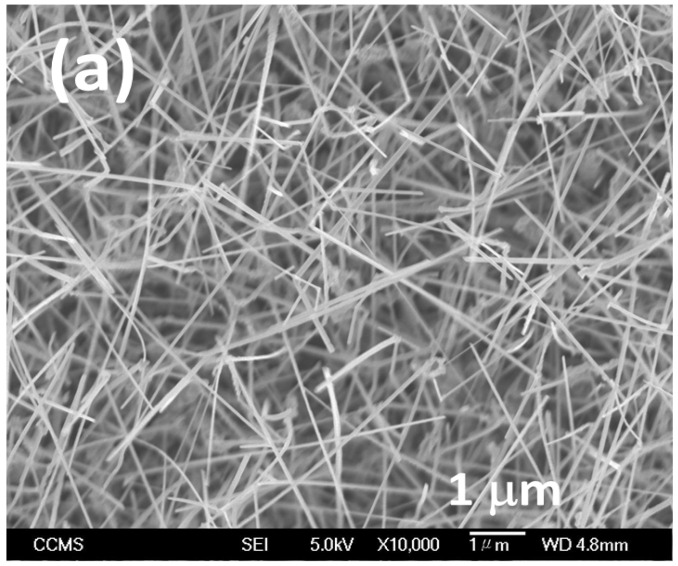
Morphology of as-grown GaN NWs: (**a**) ×10 K and (**b**) ×100 K, and (**c**) TRIM simulation for cobalt (72 keV)-doped GaN Nanowires with 1,000,000 atoms.

### 3.2. Optimization of Annealing Temperature from Photoluminescence and SEM

After ion implantation, annealing is commonly used to recover the crystalline structure. In this study, we used an RTA furnace. Four RTA temperatures of 500 °C, 600 °C, 700 °C, and 800 °C were used in NH_3_ ambient for 6 min each. Photoluminescence (PL) intensity was increased with temperature from 500 °C to 800 °C, as shown at Figure 2a.

Photoluminescence (PL) characteristic spectra of Co-doped GaN nanowires or related NWs have been studied recently [30,34,35]. M.U. Farooq et al. reported [30] that a strong band-edge emission centered at 364.5 nm (3.402 eV) was observed for undoped GaN NWs. Extra peaks were found at 376.7 nm (3.29 eV) and 383.5 nm (3.22 eV), assigned to bound magnetic polaron (BMP) and exciton magnetic polaron (EMP), respectively. They also studied [35] that Co-GaN NW has an emission peak at 363.9 nm (3.408 eV), corresponding to band-to-band edge transition of GaN, with a shoulder luminescence band at lower energy (382.9 nm, 3.24 eV). A peak at 421.1 nm (2.94 eV) was observed, assigned as a d-d transition, related to ^4^T_1_(H) → ^4^A_2_(F) of the isolated Co^2+^ ion [35], confirming the presence of Co ions in the GaN lattice. M. Maraj, et al. reported on Co-doped NWs [34], with the band-edge (BE) emission at 368.85 nm (3.361 eV) and 370.26 nm (3.348) for 6% and 8% cobalt doping, respectively. An additional peak at 455 nm (2.725 eV) was ascribed to the nitrogen vacancies in GaN.

Correspondingly, we carefully analyzed our PL spectra in Figure 2a for Co-doped GaN NWs with different RTA annealing temperatures, 500–800 °C. Under lower annealing temperatures of 500–600 °C, the strongest PL bands at near 2.9 eV were observed, and assigned as a d-d transition, related to ^4^T_1_(H) → ^4^A_2_(F) of isolated Co^2+^ ion [35], confirming the presence of Co ions in the GaN lattice. Weaker bands near 3.43 eV were seen, from the cross-band gap emission of GaN. With higher annealing temperatures of 700–800 °C, the dominant peaks appeared at ~3.33 eV, due to band-edge (BE) emission [34], accompanying with a peak at ~3.43 eV from cross-gap emission of GaN. Our PL results have revealed the effects of optimized annealing temperatures of 700–800 °C.

From SEM examination, it was observed that surface damage increased with temperature. Figure 2b shows SEM morphology on the Co-GaN NW/Si sample with dose concentration 4 × 10^16^ cm^−2^ and annealed at highest temperature (800 °C); the nanowires were damaged and bent. Figure 2c shows SEM morphology of the sample annealed at the second highest temperature (700 °C), in which the nanowires are better structured. A temperature of 700 °C was chosen as the optimizing annealing temperature.

### 3.3. X-ray Diffraction Analysis

X-ray diffraction (XRD) measurements were performed on as-grown GaN NWs and Co-doped GaN NWs before and after annealing, simply noted as as-grown, before annealing, and after annealing, respectively, in Figure 3. From a wide range scan (Figure 3a), there are six major peaks of GaN (100), (002), (101), (102), (110), and (103) because of the NW structures. Compared with the as-grown GaN NWs, the Co-implanted ones before annealing had their peaks weaker and broader, while after annealing, the annealed Co-doped GaN NW sample recovered its six peaks similar to the as-grown one. A previous investigation for Co-doping in GaN predicted some extra peaks at 65° and 44°, corresponding to CoGa (200) and CoGa (110), or hcp-Co (0002), respectively [45]. There are no such peaks observed in Figure 3a, so these compounds did not exist in our Co-GaN NWs. XRD shows a peak shift to lower angles and has short range distortions [46,47]. 

Narrow XRD scans were performed on as-grown GaN NWs and Co-doped GaN NWs before and after annealing. Figure 3b exhibits the comparative narrow scans of the GaN (002) plan for as-grown GaN NWs and for Co-implanted GaN NWs with a dose concentration of 4 × 10^16^ cm^−2^ and annealed at 700 °C. Undoped as-grown GaN NWs show the GaN (002) 2θ peak at ~34.7°. After ion implantation, it shifts to ~34.5°, and with the intensity decreased to ~70%. After annealing at 700 °C, it shifts back to the position as was observed in as-grown GaN NWs and increases its intensity over ~30% in comparison with the as-grown sample.

Figure 3c shows the comparative narrow scans of the GaN (110) plan on the same three samples of as-grown GaN NWs, Co-implanted GaN NWs, and ones annealed at 700 °C. Undoped as-grown GaN NWs have the GaN (110) 2θ peak at near 57.8°. After ion implantation, it shifts to 57.5°, but the shift is recovered after annealing. Experimental results from Figure 3b,c clearly indicate the lattice constant extended after ion implantation, with the peak intensity increased a little in comparison with the as-grown case. These observations indicate that the ion implantation damaged the crystalline structure in some way and the proper annealing at the optimized conditions recovered and improved the crystalline perfection along both the (002) and (110) plane for Co-doped GaN NWs.

All above XRD peaks were fitted by the Lorentz model for calculation. For GaN nanowires, a = 3.182 Å and c = 5.178 Å are two axes of the lattice constants. Figure 3d shows the calculated lattice constants of a and c for as-grown GaN NWs and Co-doped GaN NWs before annealing and after annealing. The a-axis from our sample expanded after ion implantation and shifted to low angle of about 0.284%. After the annealing process, the peak shifted back to a higher angle. For the lattice constant c fitting, c increased about 0.173% after ion implantation and shifted back to peak position as was observed in as-grown GaN NWs. Ion implantation usually affects the lattice constant, but the destructed lattice structure was recovered with annealing. 

### 3.4. High-Resolution Transmission Electron Microscope and Energy Dispersive X-ray Spectroscope Analysis

High-Resolution transmission electron microscope (HR-TEM) was used to investigate the structural characteristics. Figure 4 exhibits experimental results for three samples of [A] the as-grown GaN NWs, [B] the cobalt implanted GaN NWs, and [C] the sample after annealing with (a1–a3) for low magnification TEM images, with (b1–b3) for high magnification TEM images, with (c1–c3) for select area diffraction pattern (SADP), and (d1–d3) for EDX spectra.

From Figure 4A for the as-grown GaN nanowires, the SADP at (c1) indicates wurtzite structure with high crystallinity. The TEM image shows a series of SADP spots, each spot at (c1) revealing a satisfied diffraction condition of the sample’s crystal structure. Lattice-resolved view shows a highly crystalline (100) planes along the wire axis. The nanowire was observed to have a triangle-like shape and a diameter of about 80–100 nm. EDX spectrum at (d1) exhibits the elements existed in the as-grown Co-GaN NWs. Gallium and nitrogen are clearly observed in the GaN NWs as expected. 

Figure 4B shows TEM images of Co-implanted GaN NWs sample with the highest dose concentration of 4 × 10^16^ cm^−2^ at low magnification (a2) and high magnification (b2). Slight defects caused by ion implantation were observed, especially on the surface of the nanowire. These defects seem to have different directionalities. SADP at (c2) shows wurtzite structure after ion implantation. From EDX spectral analysis at (d2), a clear peak indicating Co after implantation is seen. Although defects were generated in the ion implantation process, the structure remained wurtzite.

TEM images of Co-GaN NWs after annealing revealed interesting features, in Figure 4C. Amorphous structure on the surface was observed from high magnification image of (b3). SADP at (c3) showed extra spots indicating cubic structure which were not present in Co-GaN NWs before annealing in Figure 4B (c2). The EDX analysis at (d3) displays an almost similar result before and after annealing (Figure 4B (d2)).

From HR-TEM, no clusters were observed in the nanowires. The magnetic properties to be discussed in next Section 3.3, if any, may not be related to secondary phases.

### 3.5. Magnetic Properties

The dilute magnetic properties of Co-GaN NWs are investigated by the superconducting quantum interference device (SQUID) technique. The SQUID is a technique highly sensitive to weak magnetic fields [48]. It has been used to study the Cu-doped In_x_Ga_1−x_N NWs at RT and the effects of annealing [49], GaN magnetic high electron mobility transistors (MagHEMTs) [50], the ferromagnetism in GaN doped with Cu and Mn [51], and the SQUID magnetometry of (Ga,Mn)N thin films and Mg-doped GaN [52]. For the current work, the SQUID technique was used for investigation of the magnetization. The resolution of the SQUID setup is about 10^−8^ emu (electromagnetic unit), suitable for detecting small and dilute magnetic signals. Samples with size of 5 × 5 mm^2^ were prepared and placed in the plastic tube.

Background signal from the silicon substrate was checked, and the pure diamagnetic signal of M-H (magnetization M vs. magnetic field H) was observed as shown in Figure 5a.

Figure 5b presents the measured M-H hysteresis at lowest (5 K) and room temperature (300 K) with applied field of 0–100 Oe (Oersted) on a cobalt-doped GaN Nanowire with dose concentration of 4 × 10^16^ cm^−2^ before annealing. After analysis of the results, it is found that the sample became diamagnetic with a small ferromagnetic behavior at 5 K, and only showed diamagnetic behavior at 300 K. After the silicon background was removed, paramagnetism overlapped with ferromagnetism at 300 K was observed

The sample annealed at 700 °C in NH_3_ was measured. Clear ferromagnetism was seen at 5 K and 300 K (Figure 6a) with varying hysteresis patterns. The Hc (coercivity) was 502.5 Oe at 5 K, and 201.3 Oe at 300 K.

The M–T relationship for the annealed Co-GaN NW with dose concentration of 4 × 10^16^ cm^−2^ was analyzed for determining the Curie temperature using the SQUID. A 100 Oe was applied for the zero-field cooling and field cooling (Figure 6b). The result shows the transition point between the paramagnetic property and ferromagnetic property at 332 K for the sample after annealing.

Overall, the SQUID was used to investigate the dilute magnetic properties and to check whether the ferromagnetic in Co-NWs is achieved or not by the cooling processes in the weak magnetic field. It is confirmed that the weak intrinsic magnetic properties can be maintained near room temperature.

## 4. Conclusions

Gallium nitride nanowires have been successfully grown on a Si substrate by atmospheric pressure chemical vapor deposition (APCVD) at room temperature. Samples were implanted with cobalt magnetic atoms using ion implantation and annealed by RTA at 700 °C for 6 min at NH_3_ ambience. A variety of characterization techniques, including SEM, photoluminescence (PL), XRD, HR-TEM, EDX, and superconducting quantum interference device (SQUID), were employed for characterization measurements and analyses, from which the single crystalline nature of cobalt-doped gallium nitride (Co-GaN) nanowires (NWs) was identified. 

Surface morphology from SEM of Co-doped GaN nanowires shows the wires bent and the surface was damaged. XRD scans show no secondary phase formation after ion implantation. The lattice was extended and distorted in the short-range scale. This phenomenon may be induced by stress within the nanowires. After thermal annealing, the structure shows recrystallization from XRD data, and defects such as interstitials were recovered after the annealing process. TEM image and EDX results show no clusters but EDX atomic analysis on the surface shows slight structural defects from the cobalt implanted into the nanowires. From SAEDP, the wurtzite structure was still maintained. 

The SQUID measurements showed M--H curves with hysteresis at 5 K and 300 K after the implantation process. Magnetic measurements show that Co-doped GaN NWs have room temperature ferromagnetic properties. These results are significant in the understanding of Co-doped GaN NWs and will be a useful reference in the next developments of GaN for optoelectronic, spintronic, electronic, sensing, optical, and related applications.

## Figures and Tables

**Figure 2 materials-16-00097-f002:**
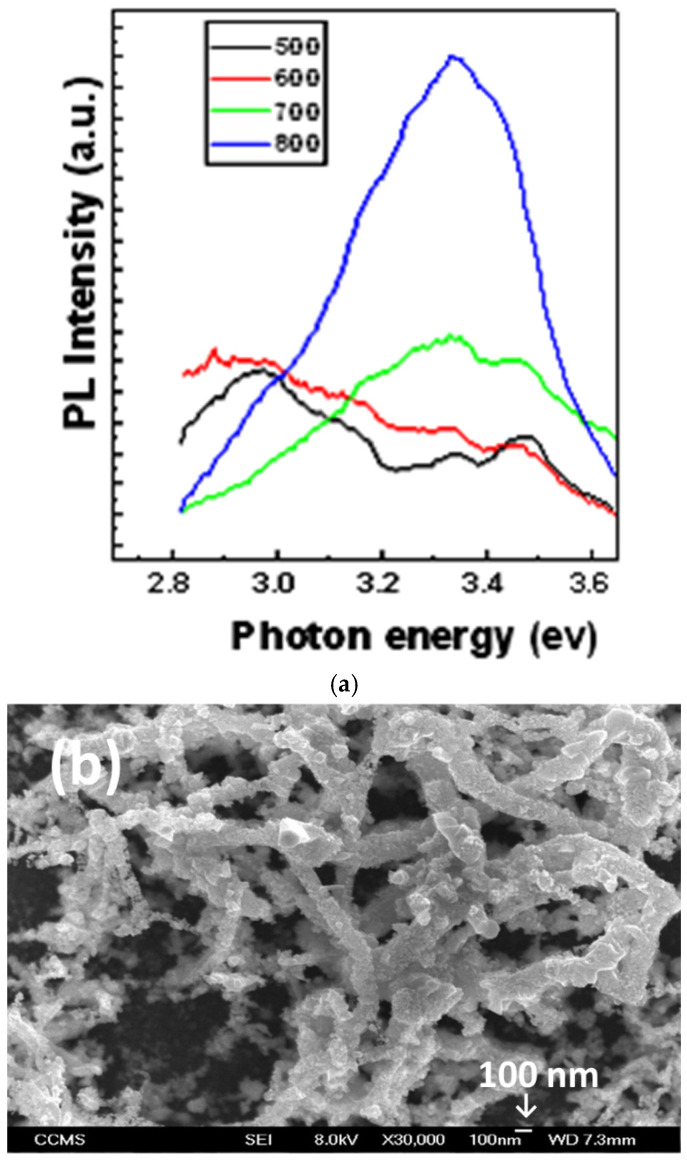
(**a**) Photoluminescence spectra (RT, 325 nm excitation) of a Co-GaN NW/Si sample with dose concentration 4 × 10^16^ cm^−2^ and annealed at different temperatures of 500, 600, 700, and 800 °C; SEM images from the Co-GaN NW/Si sample annealed at temperatures of (**b**) 800 °C and (**c**) 700 °C.

**Figure 3 materials-16-00097-f003:**
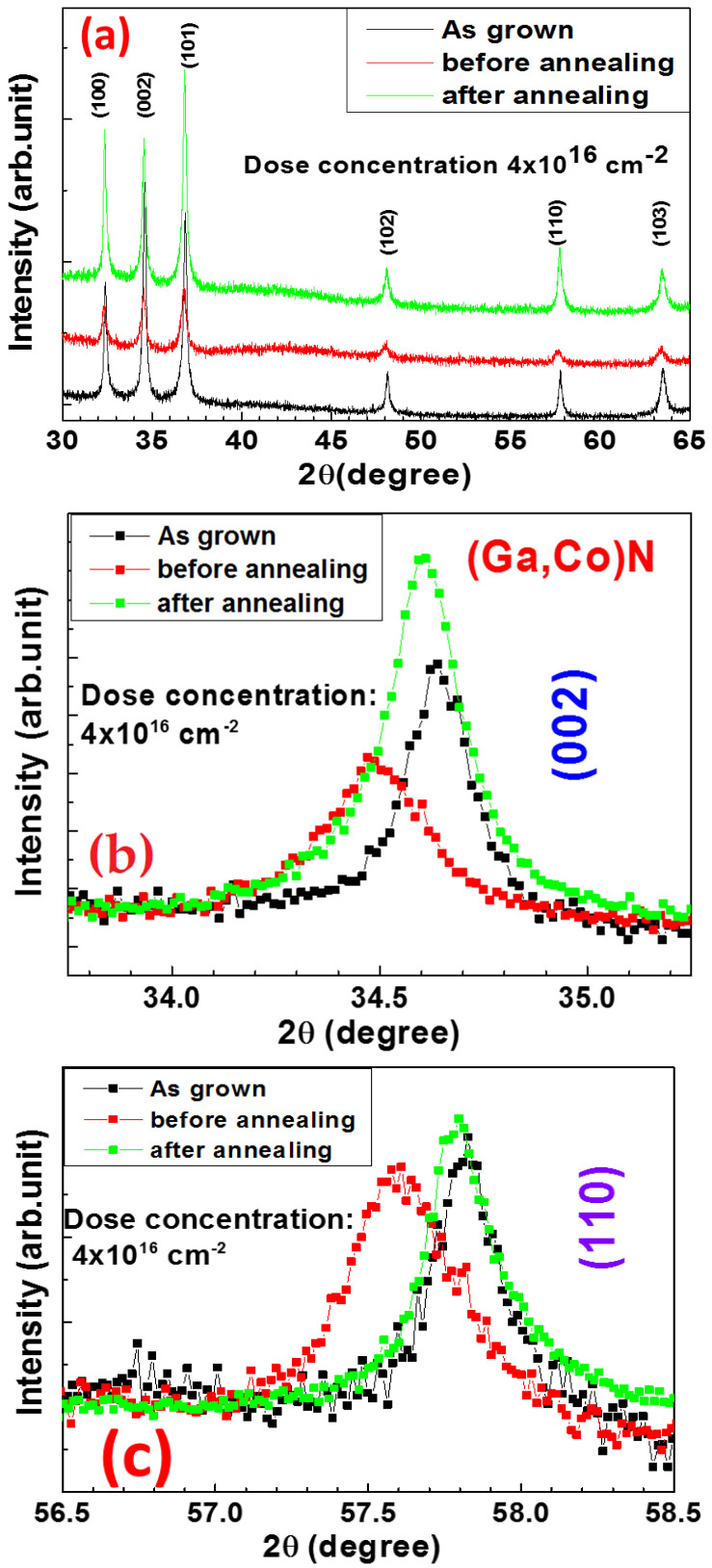
XRD scans of as-grown GaN NW and Co-GaN NW before annealing and after annealing with (**a**) wide range, (**b**) narrow range (002), and (**c**) narrow range (110); (**d**) Calculated lattice constants of a and c for these three samples.

**Figure 4 materials-16-00097-f004:**
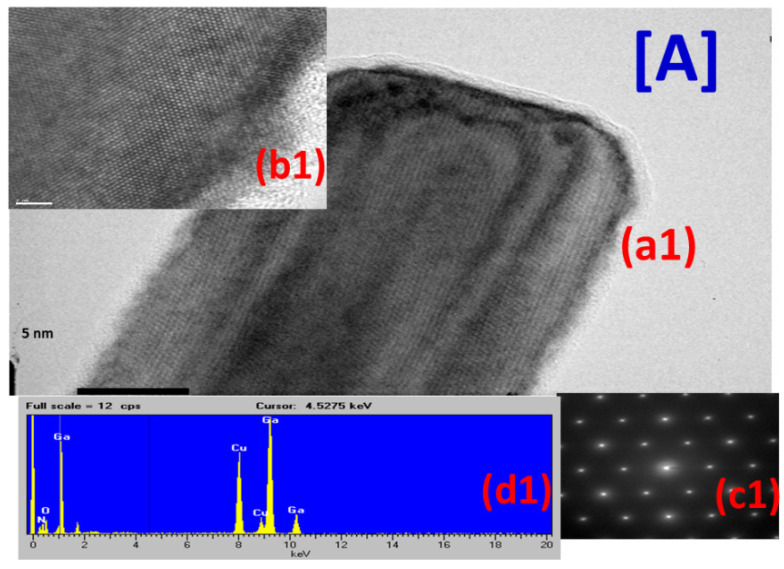
HR-TEM (High-Resolution transmission Electron Microscope) data for three samples of (**A**) as-grown GaN NWs, (**B**) cobalt implanted GaN NWs, and (**C**) annealed Co-GaN NWs; (**a1**–**a3**) show low magnification TEM images, (**b1**–**b3**) show high magnification TEM images, (**c1**–**c3**) for SADP, and (**d1**–**d3**) show EDX data.

**Figure 5 materials-16-00097-f005:**
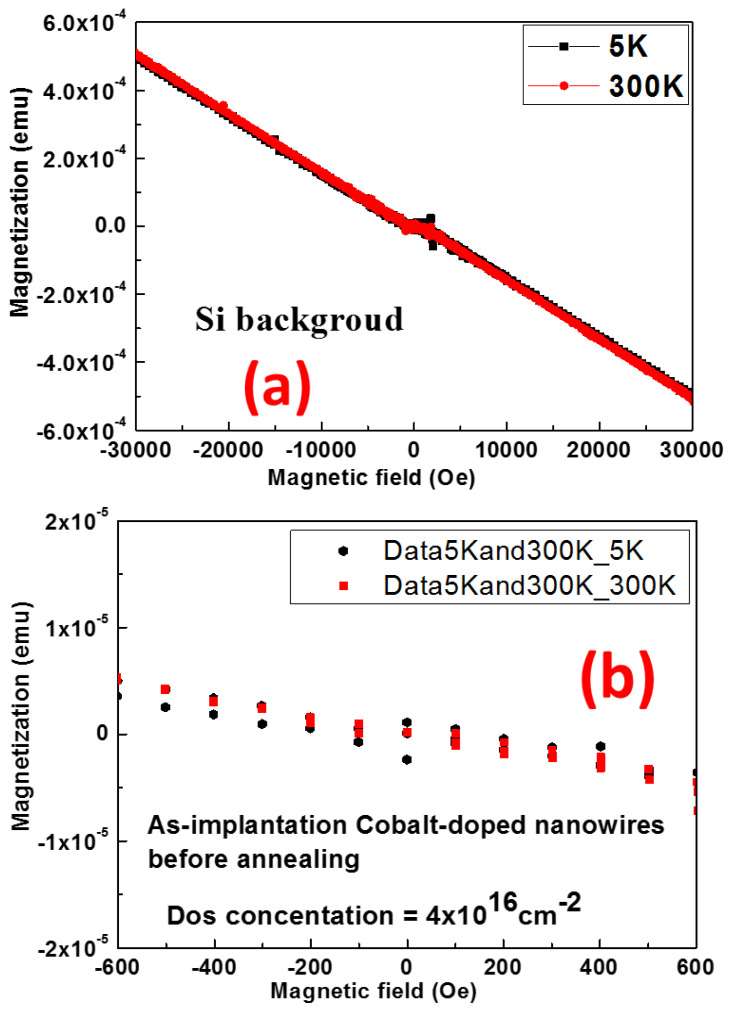
(**a**) M–H measurement from Si background at 5 K and 300 K, (**b**) M–T measurement with applied field of 0–100 Oe from a cobalt-doped GaN Nanowire before annealing with dose concentration of 4 × 10^16^ cm^−2^.

**Figure 6 materials-16-00097-f006:**
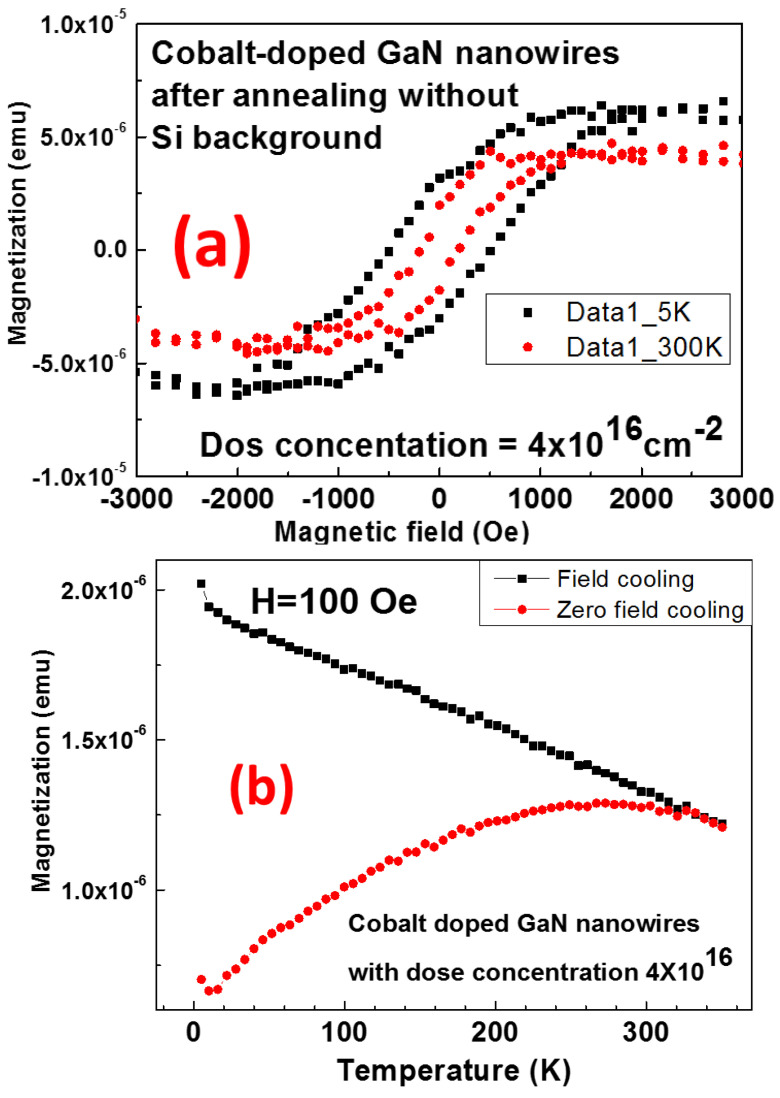
For cobalt-implanted GaN Nanowire with dose concentration of 4 × 10^16^ cm^−2^ after annealing, (**a**) M–H measurement without Si background at 5 K/300 K and (**b**) M–T measurement at applied field of 100 Oe.

**Table 1 materials-16-00097-t001:** APCVD growth key parameters.

Parameters	Values
Carrier Flow Rate	80 sccm
Carrier gas	Argon
NH_3_ Flow Rate	10 sccm
Growth Temperature	920 °C
Growth Time	1 h
Growth Pressure	760 torr
Switching Temperature	920 °C

## Data Availability

The data that support the findings of this study are available from the corresponding author: Z.C.F. (Zhe Chuan Feng), upon request.

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
