# Peer review of "Synthesis, Structural and Magnetic Properties of Cobalt-Doped GaN Nanowires on Si by Atmospheric Pressure Chemical Vapor Deposition"

_materials, 2022, doi:10.3390/ma16010097_

Round 1

Reviewer 1 Report

GaN NWs are promising candidates for photocatalytic devices. The paper presents an extensive study of their preparation using APCVD, and the conclusions are interesting and can be used in electronic, photonic and other sensing fields. The APCVD growth parameters on page 6 can be represented in the table. Did the intensity change in Figure 3(a) before and after growth, annealing, and annealing, and for what reason, the authors did not explain clearly. The paper was revised with minor revisions and recommended for acceptance. 

Author Response

Reviewer #1:

GaN NWs are promising candidates for photocatalytic devices. The paper presents an extensive study of their preparation using APCVD, and the conclusions are interesting and can be used in electronic, photonic and other sensing fields. The APCVD growth parameters on page 6 can be represented in the table. Did the intensity change in Figure 3(a) before and after growth, annealing, and annealing, and for what reason, the authors did not explain clearly. The paper was revised with minor revisions and recommended for acceptance. 

Response: Thanks for the constructive opinions. We’ve made a Table 1 APCVD growth parameters, as below,

Table 1 APCVD growth parameters

parameters

values

Carrier Flow Rate             

80 sccm

Carrier gas

Argon

NH3 Flow Rate

10 sccm

Growth Temperature

920℃

Growth Time

1 hour

Growth Pressure

760 torr

Switching Temperature

920℃

For the XRD “intensity change in Figure 3(a) before and after growth, annealing, and annealing, and for what reason”, we have discussed in Section 3.3. X-ray diffraction analysis, especially related to Figures 3 (a), (b) and (c). We’ve further modified the 2nd and 3rd paragraphs, as below,

Figure 3 (b) exhibits the comparative narrow scans of GaN (002) plan for as-grown GaN NW and for Co-implanted GaN NW with a dose concentration of 4×1016 cm-2 and annealed at 700oC. Undoped as-grown GaN NW shows the GaN (002) 2q peak at ~34.7o. After ion implantation, it shifts to ~34.5o, and with the intensity decreased to ~70%. After 700°C annealing, it shifts back to the position as was observed in as-grown GaN NWs and increases its intensity over ~30% in comparison with the as-grown sample.

Figure 3 (c) shows the comparative narrow scans of GaN (110) plan on the same three samples of as-grown GaN NW, Co-implanted GaN NW and annealed one at 700oC. Undoped as-grown GaN NW has the GaN (110) 2q peak at near 57.8o. After ion implantation, it shifts to 57.5o, with the peak intensity decreased a little, but the shift is recovered after annealing, with the peak intensity increased a little, in comparison with the as-grown case. These observations indicate that the ion implantation damaged the crystalline structure in some content and the proper annealing at the optimized condition recovered and improved the crystalline perfection along both the (002) and (110) plane for Co-doped GaN NWs.

Also, in the last paragraph of this Section 3.3, we commended on Fig. 4 (d) and discussed the lattice constants of a and c for as-grown GaN NWs, Co-doped GaN NWs before annealing and after annealing, respectively, which were calculated from the XRD data of Figs. 4 (b) and (c). We demonstrate: “The a-axis from our sample expanded after ion implantation and shifted to low angle of about 0.284%. After the annealing process, the peak shifted back to a higher angle. For the lattice constant c fitting, c increased about 0.173% after ion implantation and shifted back to peak position as was observed in as-grown GaN NWs. Ion implantation usually affects the lattice constant, but the destructed lattice structure was recovered with annealing.”. In the abstract, we’ve mentioned “Slight expansion in the lattice constant of Co-GaN NWs due to the implantation-induced stress effect was observed, which was recovered by thermal annealing”.

Reviewer 2 Report

Please see attached word document.

Author Response

Reviewer#2:

1) Can the authors increase the quality for the photoluminescent (PL) plot in Fig. 2(a)? Right now, the y axis tick values are missing, and the legend is not clear.

Response: Thanks for the good comments. We’ve input a better-quality PL Fig. 2(a), into the manuscript, as below,

Previous one is of poor quality due to that we input the graph into Power Point to add (a) into the graph, which lowered the quality.

2) Does the Co doping change the band gap of GaN nanowires? Has the authors notice any differences or changes from the photoluminescent results (for example peak position) for the Co doped NWs?

Response: Thanks for the constructive comments. We’ve added two paragraphs after the 1st paragraph of Section 3.2:

Photoluminescence (PL) characteristic spectra of Co-doped GaN nanowires or related are studied recently [30,34,35]. M.U. Farooq et al. reported on Cr-doped GaN NWs [30] that a strong band-edge emission centered at 364.5 nm (3.402-eV) was found for undoped GaN NWs. Extra peaks were observed at 376.7 nm (3.29 eV) and 383.5 nm (3.22 eV), assigned to bound magnetic polaron (BMP) and exciton magnetic polaron (EMP). They also studied [35] that Co-doped GaN NW has emission peak at 363.90 nm (3.408 eV), correspond to band-to-band edge transition of GaN, with a shoulder luminescence band at lower energy (382.9 nm, 3.24 eV). A peak at 421.1 nm (2.94 eV) was observed, assigned as a d-d transition, related to 4T1(H) → 4A2(F) of isolated Co2+ ion, confirming the presence of Co ions in the GaN lattice. M. Maraj, et al. reported on Co-doped NWs [34], that band-edge (BE) emission at 368.85 nm (3.361 eV) for 6% and 370.26 nm (3.348) for 8% cobalt doping, causing a small red shift due to the inclusion of cobalt in GaN. This type of minor red shifting in band edge peak was mostly attributed to the strains occurring because of Co and other impurity (defects) in the GaN structures. A peak at 455 nm (2,73 eV) was ascribed to the nitrogen vacancies in GaN that may also arise due to strain and defects in the GaN structures. PL, along with the near band edge emission at 367 nm (3.376-eV), a relatively weaker peak in the blue emission region is ascribed to the nitrogen vacancies. A smaller shift in the peak position is attributed to the presence of Co, giving rise to the strain shifting of peaks.

Correspondingly, we carefully analyze our PL spectra in Fig. 2 (a) for Co-doped GaN NWs with different RTA annealing temperatures, 500-800oC. Under lower annealing temperatures of 500-600oC, strongest PL bands at near 2.9 eV were observed, and assigned as a d-d transition, related to 4T1(H) → 4A2(F) of isolated Co2+ ion, confirming the presence of Co ions in the GaN lattice [35]. Weaker bands near 3.43 eV were seen, from the cross-band gap emission of GaN. With higher annealing temperatures of 700-800oC, the dominant peaks were appeared at ~3.33 eV, due to the band-edge (BE) emission [34], accompanying with a should peak at ~3.43 eV from the cross-band gap emission of GaN. Our PL results have revealed the effects of optimized annealing temperatures of 700-800oC.

3) In the EDX data in figure 4, a clear peak related to Cu is present (EDX data in Fig 4 (d1)), can the author explain the reason for that?

Response: Thanks for the comments. Cu is present (EDX data in Fig 4 (d1) could be due to the use of Cu-target in the XRD and EDX measurements.

4) For the magnetic property, can the authors make a plot to show the magnetic susceptibility for the Co doped NWs?

Response: Thanks for the recommendation. Sorry, we didn’t do measurements on the magnetic susceptibility for our samples and unable to add. Indeed, we’ve searched via Google search for GaN (and ZnO) nanowires magnetic susceptibility but can’t find any reference articles in the literature.

5) In Fig. 6(a), the M-H hysteresis loop related to the ferromagnetic state is shown. Furthermore, a transition from paramagnetic to ferromagnetic is observed under T=332K. On top of these, can the authors show the magnetization vs T at a fixed magnetic field for the ferromagnetic component? Can the authors also comment or show the magnetic susceptibility for the Co doped NWs?

Response: Thanks for the valuable comments. We have considered these opinions carefully and thoroughly. But, extremely sorry, we are unable now to add or make the magnetization vs T at a fixed magnetic field for the ferromagnetic component. On the magnetic susceptibility for the Co doped NWs, to my knowledge, magnetic susceptibility is an important parameter for magnetic materials. However, GaN (and ZnO) nanowires, even can show some dilute magnetic properties as we studied in the current article, are still unable to treat as standard magnetic materials. As I searched for “dilute magnetic semiconductor magnetic susceptibility”, more reference papers can be found, mostly for II-VI (such as CdMnTe, ZnMnSe, etc), but none for GaN (and ZnO) nanowires.

              In the Introduction, lines 112-119, we mentioned: “GaN NWs doped with 3d-transition metals ions (Mn, Fe, Cr, Co, and Ni) are interesting dilute magnetic semiconductors (DMSs). Partially filled d-states contain unpaired electrons that can introduce spin properties in GaN NWs. This spin due to the hybridization of the magnetic impurity with s- and p-states of the host semiconductor, is beneficial for spintronic based applications. Upon comparison of transition metal (TM) doped with other III-V semiconductors such as GaAs, InAs and InP, GaN exhibits Curie temperature (Tc) above RT. However, the origin of ferromagnetism in these semiconductors is still controversial and not well understood [30].”

In this paper, we are focusing and employing the superconducting quantum interference device (SQUID) technique to investigate the magnetic properties of Cobalt-doped GaN nanowires under relatively low magnetic field. To enhance this focusing work, we’ve added five references [48-52] to introduce more on SQUID and modified the 1st paragraph of Section 3.5. Magnetic properties, before Figure 5, as below:

The magnetic properties of Co-doped GaN NWs are investigated by the superconducting quantum interference device (SQUID) technique. SQUID is known to be extremely sensitive to weak magnetic fields [48] and has been used to study the magnetic InxGa1xN nanowires at room temperature using Cu dopant and annealing [49], GaN magnetic high electron mobility transistors (MagHEMTs) [50], hole‑mediated ferromagnetism in GaN doped with Cu and Mn [51], and the improved-sensitivity integral SQUID magnetometry of (Ga,Mn)N thin films in proximity to Mg-doped GaN [52]. In the current work, SQUID technique was used for investigation of the magnetization. The resolution of the SQUID setup is about 10-8 emu (electromagnetic unit), suitable for detecting small and dilute magnetic signals. Samples with size of 5×5 mm2 were prepared and placed in the plastic tube.

Also, two sentences in the end of Section 3.5. are added to briefly review the function of the superconducting quantum interference device (SQUID), as:

Overall, the SQUID was used to investigate the dilute magnetic properties and to check whether the ferromagnetic in Co-NWs is achieved or not by the cooling processes in the weak magnetic field. It is confirmed that the weak intrinsic magnetic properties can be maintained near room temperature.  

Reviewer 3 Report

The manuscript by Feng et al. represents an extremely detailed description of work on the fabrication and characterization of cobalt-doped GaN nanowires. After a very good introduction, the experimental work apparently performed by the authors on crystal growth and further processing is documented. A number of characterization methods are then applied and these results are also presented. This brief description given here by the reviewer is also the content of the Conclusion section.

Overall, apart from an activity report, which may well contain interesting information for a reader planning similar work, we see no scientific findings generalizable in any way to materials science at all. Unfortunately, the authors also make it difficult for the reviewer, if there is something really new e.g. on growth mechanisms, to find it even in the long text full of details.

In our opinion, this paper is at best a project- or activity report, but not a scientific paper. It would have been the authors' responsibility to gain scientific knowledge from the material. To be clear: We think the topic of the manuscript is interesting and important. However, a publication must be more than just a description of work that has been done.

Author Response

Reviewer #3 comments:

Overall, apart from an activity report, which may well contain interesting information for a reader planning similar work, we see no scientific findings generalizable in any way to materials science at all. Unfortunately, the authors also make it difficult for the reviewer, if there is something really new e.g. on growth mechanisms, to find it even in the long text full of details.

In our opinion, this paper is at best a project- or activity report, but not a scientific paper. It would have been the authors' responsibility to gain scientific knowledge from the material. To be clear: We think the topic of the manuscript is interesting and important. However, a publication must be more than just a description of work that has been done.

Response: Thanks for the valuable comments. We are making efforts to make new revised manuscript and responses, to try to enhance our scientific observations and understandings, to gain scientific knowledge on Co-doped GaN nanowires, at least in some contents.

[A] We’ve added 2-paragraphes in Section 3.2 to further analyze our measured PL spectra of Co-doped GaN NWs under RTA annealing with different temperature of 500-800oC, with some more scientific discussions as below:

Photoluminescence (PL) characteristic spectra of Co-doped GaN nanowires or related are studied recently [30,34,35]. M.U. Farooq et al. reported on Cr-doped GaN NWs [30] that a strong band-edge emission centered at 364.5 nm (3.402-eV) was found for undoped GaN NWs. Extra peaks were observed at 376.7 nm (3.29 eV) and 383.5 nm (3.22 eV), assigned to bound magnetic polaron (BMP) and exciton magnetic polaron (EMP). They also studied [35] that Co-doped GaN NW has emission peak at 363.90 nm (3.408 eV), correspond to band-to-band edge transition of GaN, with a shoulder luminescence band at lower energy (382.9 nm, 3.24 eV). A peak at 421.1 nm (2.94 eV) was observed, assigned as a d-d transition, related to 4T1(H) → 4A2(F) of isolated Co2+ ion, confirming the presence of Co ions in the GaN lattice. M. Maraj, et al. reported on Co-doped NWs [34], that band-edge (BE) emission at 368.85 nm (3.361 eV) for 6% and 370.26 nm (3.348) for 8% cobalt doping, causing a small red shift due to the inclusion of cobalt in GaN. This type of minor red shifting in band edge peak was mostly attributed to the strains occurring because of Co and other impurity (defects) in the GaN structures. A peak at 455 nm (2,73 eV) was ascribed to the nitrogen vacancies in GaN that may also arise due to strain and defects in the GaN structures. PL, along with the near band edge emission at 367 nm (3.376-eV), a relatively weaker peak in the blue emission region is ascribed to the nitrogen vacancies. A smaller shift in the peak position is attributed to the presence of Co, giving rise to the strain shifting of peaks.

Correspondingly, we carefully analyze our PL spectra in Fig. 2 (a) for Co-doped GaN NWs with different RTA annealing temperatures, 500-800oC. Under lower annealing temperatures of 500-600oC, strongest PL bands at near 2.9 eV were observed, and assigned as a d-d transition, related to 4T1(H) → 4A2(F) of isolated Co2+ ion, confirming the presence of Co ions in the GaN lattice [35]. Weaker bands near 3.43 eV were seen, from the cross-band gap emission of GaN. With higher annealing temperatures of 700-800oC, the dominant peaks were appeared at ~3.33 eV, due to the band-edge (BE) emission [34], accompanying with a should peak at ~3.43 eV from the cross-band gap emission of GaN. Our PL results have revealed the effects of optimized annealing temperatures of 700-800oC.

[B] In Section 3.3. X-ray diffraction analysis, especially related to Figures 3 (a), (b) and (c), we have discussed further and modified the 2nd and 3rd paragraphs, as below:

Figure 3 (b) exhibits the comparative narrow scans of GaN (002) plan for as-grown GaN NW and for Co-implanted GaN NW with a dose concentration of 4×1016 cm-2 and annealed at 700oC. Undoped as-grown GaN NW shows the GaN (002) 2q peak at ~34.7o. After ion implantation, it shifts to ~34.5o, and with the intensity decreased to ~70%. After 700°C annealing, it shifts back to the position as was observed in as-grown GaN NWs and increases its intensity over ~30% in comparison with the as-grown sample.

Figure 3 (c) shows the comparative narrow scans of GaN (110) plan on the same three samples of as-grown GaN NW, Co-implanted GaN NW and annealed one at 700oC. Undoped as-grown GaN NW has the GaN (110) 2q peak at near 57.8o. After ion implantation, it shifts to 57.5o, with the peak intensity decreased a little, but the shift is recovered after annealing, with the peak intensity increased a little, in comparison with the as-grown case. These observations indicate that the ion implantation damaged the crystalline structure in some content and the proper annealing at the optimized condition recovered and improved the crystalline perfection along both the (002) and (110) plane for Co-doped GaN NWs.

In the last paragraph of this Section 3.3, we commended on Fig. 4 (d) and discussed the lattice constants of a and c for as-grown GaN NWs, Co-doped GaN NWs before annealing and after annealing, respectively, which were calculated from the XRD data of Figs. 4 (b) and (c). We demonstrate: “The a-axis from our sample expanded after ion implantation and shifted to low angle of about 0.284%. After the annealing process, the peak shifted back to a higher angle. For the lattice constant c fitting, c increased about 0.173% after ion implantation and shifted back to peak position as was observed in as-grown GaN NWs. Ion implantation usually affects the lattice constant, but the destructed lattice structure was recovered with annealing.”. In the abstract, we’ve mentioned “Slight expansion in the lattice constant of Co-GaN NWs due to the implantation-induced stress effect was observed, which was recovered by thermal annealing”.

[C] In the Section 3.5. Magnetic properties, to enhance our scientific discussion on the superconducting quantum interference device (SQUID), we’ve added five references [48-52], modified the 1st paragraph and added two sentences in the end of text in this section, as below:  

            To my knowledge, no reports on using SQUID for investigation Co-doped GaN NWs are found yet, except of Ref. [49] on using SQUID to study Cu-doped InxGa1xN nanowires.

[D] In Abstract, we’ve briefly concluded our main scientific/technological contributions: “Co-doped GaN showed optimum structural properties when annealed at 700 oC for 6 minutes in NH3 ambience. From scanning electron microscopy (SEM), x-ray diffraction (XRD), high resolution transmission electron microscope (HRTEM) and energy dispersive X-ray spectroscopy (EDX) measurements and analyses, single crystalline nature of Co-GaN NWs was identified. Slight expansion in the lattice constant of Co-GaN NWs due to the implantation-induced stress effect was observed, which was recovered by thermal annealing. Co-GaN NWs exhibited ferromagnetism as per superconducting quantum interference device (SQUID) measurement. Hysteretic curves with Hc (coercivity) of 502.5 Oe at 5K and 201.3 Oe at 300K, respectively, were observed. With an applied magnetic field of 100 Oe, the transition point between the paramagnetic property and ferromagnetic property was determined at 332K. Interesting structural and conducive magnetic properties”.

Zhe Chuan Feng (corresponding author)

Round 2

Reviewer 3 Report

The manuscript has been improved.